# The Varietal Influence of Flavour Precursors from Grape Marc on Monoterpene and C_13_-Norisoprenoid Profiles in Wine as Determined by Membrane-Assisted Solvent Extraction (MASE) GC-MS

**DOI:** 10.3390/molecules27072046

**Published:** 2022-03-22

**Authors:** Lisa Pisaniello, Flynn Watson, Tracey Siebert, Leigh Francis, Josh L. Hixson

**Affiliations:** The Australian Wine Research Institute, P.O. Box 197, Glen Osmond, SA 5064, Australia; lisa.pisaniello@awri.com.au (L.P.); flynn.watson@awri.com.au (F.W.); tracey.siebert@awri.com.au (T.S.); leigh.francis@awri.com.au (L.F.)

**Keywords:** membrane-assisted solvent extraction, pomace, by-product, glycosides, hydrolysis

## Abstract

The winemaking by-product grape marc (syn. pomace) contains significant quantities of latent flavour in the form of flavour precursors which can be extracted and used to modulate the volatile composition of wine via chemical hydrolysis. Varietal differences in grapes are widely known with respect to their monoterpene content, and this work aimed to extend this knowledge into differences due to cultivar in volatiles derived from marc precursors following wine-like storage conditions. Marc extracts were produced from floral and non-floral grape lots on a laboratory-scale and from Muscat Gordo Blanco marc on a winery -scale, added to a base white wine for storage over five to six months, before being assessed using a newly developed membrane-assisted solvent extraction gas chromatography-mass spectrometry (GC-MS) method. The geraniol glucoside content of the marc extracts was higher than that of juices produced from each grape lot. In all wines with added marc extract from a floral variety, geraniol glucoside concentration increased by around 150–200%, with increases also observed for non-floral varieties. The relative volatile profile from extracts of the floral varieties was similar but had varied absolute concentrations. In summary, while varietally pure extracts would provide the greatest control over flavour outcomes when used in winemaking, aggregated marc parcels from floral cultivars may provide a mechanism to simplify the production logistics of latent flavour extracts for use in the wine sector.

## 1. Introduction

Monoterpenes play an important role in the aroma and flavour of white grapes and wine, especially for Muscat varieties and those deemed ‘floral’ due to the characteristics that monoterpenes impart, such as Gewürztraminer [1]. While free monoterpenes are present in the grapes of some varieties [2], they are mainly present as non-volatile and non-odiferous glycoconjugates [3,4]. These bound monoterpenes first require hydrolytic breakdown to release the free aglycone and impart their characteristic aromas in wine. From grapes to wine, bound monoterpenes can undergo enzymatic hydrolysis to release monoterpenes via the action of yeast, bacteria, or exogenous enzymes during winemaking [5], or through continual chemical hydrolysis as wine ages [6].

The hydrolysis of bound flavour precursors and subsequent release of volatile compounds have been of interest to wine for decades [6,7,8,9], with studies investigating extraction for the purposes of quantification or in vitro evaluation [10,11,12] and application of exogenous grape or wine-derived precursors to modulate flavour outcomes [13,14,15]. More recently, attention has focused on the flavour precursor potential in grape marc [16,17,18].

Grape marc contains flavour potential, or latent flavour, which is usually discarded but can be extracted and used to yield flavour in wine, including by in-mouth hydrolysis [17]. Not only can hydrolysis of precursors occur via key winemaking stages but can also provide a slow release of flavour over years in the bottle [19], potentially aiding in increasing shelf-life by retaining flavour. Potent monoterpenes responsible for ‘floral’ attributes such as geraniol and linalool are found at higher concentrations in young wines due to their subsequent rearrangement into less potent analogues [4], and may therefore be associated with wine youthfulness [20].

In the work of Munoz-Gonzalez et al. (2013), volatiles were liberated after enzymatic hydrolysis of Verdejo marc extract showing the potential of this waste stream [16]. However, the grape marc was first lyophilised and ground to a powder before being subjected to pressurised liquid extraction then solid-phase extraction, a process that may prove problematic with respect to scalability. Jelley and co-workers simplified the extraction process by using a crude liquid extract of Sauvignon Blanc marc in model fermentations to yield odiferous thiols [18], although the low fidelity of their approach may have implications for unwanted phenolics and bitterness [21]. The extraction outlined by Parker et al. (2019) achieved the isolation of a glycoside-rich extract using wine-allowable inputs and processes on 300 kg of Gewürztraminer marc that achieved an increase in the wine flavour without any additional bitterness related to phenolic compounds [17]. Additionally, it was noted that minimal differences were observed in outcomes between marc extracts added pre- or post-fermentation, suggesting that the main driver of flavour release was not through yeast, bacteria or enzyme action but by acid-catalysed hydrolysis. With respect to winemaking applications of a marc extract, it would allow for winemakers to make reactive additions after the outcome of fermentation is known, or possibly to modulate a product that is not necessarily reliant on fermentation such as a low- or no-alcohol product.

Studies on the hydrolysis of glycosides to yield flavour have largely focussed on quantitation of the volatiles derived from aglycone liberation to determine the outcome [14,22,23,24,25]. In some instances, the glycosides themselves have been monitored before and after a particular winemaking stage or period of storage [13,17]. In the case of Ugliano et al. (2006), fermentation with different yeasts in the presence of a glycosidic mixture isolated from grape juice produced a decline in total glycosides. Additionally, every individual glycoside monitored declined, significantly or otherwise, which was also the case in their non-inoculated control. In contrast, the work outlined by Parker et al. (2019) monitored the change in geraniol glucoside concentration after six months of storage, with the same wines analysed over a further 30 months in a subsequent report [19]. Interestingly, while not specifically highlighted, the concentration of geraniol glucoside expected in the wines based on the concentration in the extract was significantly lower than what was observed in the wines, suggesting a net formation of geraniol glucoside across six months of ageing.

The evolution of geraniol glucoside could be explained by the release from disaccharide forms (or higher) [26], which have been observed in grapes [27,28,29,30]. However, extensive glycoside profiling has not been performed during the course of winemaking, or for winemaking by-products, such as marc. Therefore, the increase in geraniol glucoside implied by the work of Parker et al. (2019) during the storage of a grape marc extract in wines requires further attention.

Furthermore, the varietal bearing on monoterpene content, composition and localisation in the grape is well established [2,31]. However, varietal changes in monoterpene localisation between the juice, pulp and skin create the possibility that grape marc parcels targeted for flavour precursor recovery differ from those with favourable monoterpene profile in juices and wines. While it is fair to assume that the monoterpene trends in juices or wines are directly transferable to grape marc precursors, it is yet to be shown. 

Here, extracts have been created from varietally pure grape marc parcels at either a winery or laboratory scale using food-grade materials and processes, and subsequently added to commercial wines and stored over several months under cellar-like conditions. The evolution of volatiles was monitored using a membrane-assisted solvent extraction gas chromatography-mass spectrometry (GC-MS) method that encompassed monoterpenes and C_13_-norisoprenoids. The aims were: to investigate the possibility of an increase in geraniol glucoside from a marc extract after several months in wine by regular monitoring over the initial months of storage; to create marc extracts from grape varieties considered floral or non-floral and explore the link between variety, monoterpene potential and flavour evolution during wine storage; and to use the developed knowledge for logistical considerations of using grape marc as a source of latent flavour.

## 2. Results

### 2.1. Validation of Analytical Methodology

In previous works, the analysis of monoterpenes and C_13_-norisoprenoids in wines has been performed using a number of discrete stable isotope dilution GC-MS analyses [17]. To consolidate the analysis of key wine aroma compounds into a single method and to continually improve analytical performance, membrane-assisted solvent extraction (MASE) was employed prior to the injection of the extract into the instrument [32,33]. Furthermore, the MASE technique allowed for simplified sample preparation and was automated by utilising a Gerstel multipurpose sampler (MPS) Robotic Pro. The calibration function was obtained in duplicate in either a commercially available Chardonnay for validation in a white wine matrix (Table 1), or commercially available Shiraz or Pinot Noir for validation in a red wine matrix (Appendix A). The MASE-GC-MS method provided good chromatographic separation of the target analytes as well as linearity for each compound and in red or white wine matrices in ranges that extended below the individual aroma detection thresholds. The limits of detection (LOD) and quantitation (LOQ) were determined from the signal to noise ratio (S/N) as three-times the S/N and ten-times the S/N, respectively. For subsequent analyses, including in the white wines used throughout this study, the LOD and LOQ were determined for each analytical batch and where relevant can be found in the Appendix A.

Each analyte was spiked into the analysis matrix in replicate (*n* = 7) at both a high and low concentration (see Materials and Methods for concentrations) to determine compound recovery and repeatability. The low concentration spikes and recoveries were between 91 and 111% except for TDN, β-damascenone and β-ionone, which were between 80 and 85%. For the high spikes, recoveries for TDN, β-damascenone and β-ionone were improved, and all compounds had recoveries between 95 and 112%.

### 2.2. Winery-Scale Muscat Gordo Marc Extraction and Addition to Wine

Muscat Gordo marc that was collected in vintage 2019 was extracted at a winery-scale using similar conditions to that previously reported for Gewürztraminer marc [17]. The Muscat Gordo marc extraction and purification yielded 145.55 g of extract from 300 kg of marc (0.485 g/kg) and subsequent analysis of bound monoterpenes by LC-MS/MS gave a geraniol glucoside concentration of 0.94 μg/mg in the extract. Geraniol glucoside made up approximately 80% of the monoterpene glucosides, based on a quantification of all other peaks with the same mass fragmentation pattern corresponding to a monoterpene aglycone molecular mass of 154.25 g/mol (geraniol, nerol, linalool, α-terpineol) bound to a hexose, in geraniol glucoside equivalents.

To determine the changes in geraniol glucoside concentration in the initial months of storage in wine, the Muscat Gordo marc extract was added to a commercial Chardonnay wine with regular monitoring of bound monoterpene and free monoterpene concentrations over 156-days of storage (Figure 1). At the addition rate of 0.404 g of extract per litre of wine, the expected geraniol glucoside concentration due to the addition was 380 μg/L. The geraniol glucoside concentration increased across the first six months of storage to 742 μg/L, or 195% of the initial concentration.

From the same LC-MS/MS analysis, the peaks for the same monoterpene mass as above but bound to either pentosyl-glucose (PGs) or rhmanosyl-glucose (RGs) were quantified. At the time of analysis, the only deuterated disaccharide internal standard available to us was syringol gentiobioside. This analysis was performed to give an indication of monoterpene disaccharide breakdown as a mechanism of releasing geraniol glucoside. In terms of relative peak areas, the monoterpene PGs were much more abundant than the monoterpene RGs, and both showed a decrease over the storage period.

As expected, the hydrolysis of Muscat Gordo marc extract in wine yielded free monoterpenes, with both geraniol and linalool above their respective theoretical sensory thresholds at the first analysis timepoint (41 days). The most abundant monoterpenes shown in Figure 1, geraniol, linalool and α-terpineol, were still increasing after 156-days of storage, with the concentration of C_13_-norisoprenoids and other monoterpenes found in the Appendix A.

While the results here confirm the increase in geraniol glucoside concentration over the first months of storage of a marc extract in wine, there are still questions on the practicality of using marc as a flavour source. Specifically, the logistics concerned with sourcing varietally pure marc versus exploiting a mixed variety white marc source at a natural aggregation point such as a processing facility or bulk collection that does not require a winery specifically having to segregate marc parcels from each press run.

### 2.3. Laboratory-Scale Varietal Marc Extraction

Several grape lots were collected that represented floral or non-floral varieties, according to expected monoterpene content. These grapes were used to understand the varietal impact of obtaining a marc extract for boosting floral aroma in winemaking. All grape samples were crushed and pressed within a few days of harvest, and the resulting marc and juice were frozen for later analysis. The grape marc samples were extracted in the laboratory using a standard phenol-free glycoside isolation procedure, a scaled-down version of that used for the winery-scale extraction of Muscat Gordo marc, above. The mass of purified and dried extract per weight of marc or grapes showed a minimal difference between floral and non-floral varieties, with the mean value of each group separated by roughly the 95% confidence interval determined by analysis of variance (Figure 2A,B). The extracts from the floral varieties showed a significantly higher geraniol glucoside concentration per mass of extract than the non-floral varieties (Figure 2C), which is expected based on the categorisation of the marc types. For these comparisons (Figure 2A–C), data for each marc type passed a Shapiro-Wilk test of normality, although most likely due to the small sample sizes (*n* = 5 or 6 per marc type). As such, the same data were subjected to Wilcoxon rank-sum tests and gave similar results. Comparisons between the groups yielded *p*-values of 0.014, 0.055 and 0.008 for extract yield per weight of marc, extract yield per weight of grapes and geraniol glucoside content, respectively. The Tukey Honest Significant Differences at 95% have been used in Figure 2A–C for visual simplicity.

The relationship between geraniol glucoside concentration in the marc extract and the geraniol glucoside concentration in the juice showed a weak correlation that was largely driven by the high concentrations in each analysis for the floral varieties (Figure 2D). While calculating the ratio between these two analyses would provide a single variable to be compared, for some varieties the concentration of geraniol glucoside in marc and/or juice was below the limit of detection and prevented a ratio from being determined. Three samples had geraniol glucoside concentrations below the limit of detection in the marc extract (Chardonnay) or juice (Riesling, Chardonnay and Cabernet Sauvignon). However, the weak correlation between juice geraniol glucoside and marc geraniol glucoside concentration suggests that an understanding of juice or wine bound monoterpene concentration may not necessarily transfer to that of a marc extract.

When the bound monoterpene analysis was expanded to include the quantification of monoterpene pentose-glucoses (PGs) and rhamnose-glucoses (RGs) in syringol gentiobioside equivalents, the compositional profiles between the marc extracts (Figure 2E) was independent of the juice profiles (Figure 2F) for most of the varieties, regardless of categorisation as either floral or non-floral. In the case of Chardonnay, no geraniol glucoside was detected in the marc extract or juice. For all others, geraniol glucoside appeared to be preferentially maintained in the grape marc when compared with the disaccharide classes (PGs or RGs), whether that be by differences in extraction or retention during processing or due to the extraction and isolation process selectively removing the higher-order monoterpene-sugar analogues and effectively enriching the geraniol glucoside concentration.

In this analysis, the disaccharides were quantified against deuterated syringol gentiobioside and the absolute concentrations obtained may not be directly comparable with the geraniol glucoside concentration. However, the relative compositional profile differences between extract and juice avoid this issue to some extent due to the relative nature of the comparison.

The marc extracts were also subjected to hydrolytic conditions followed by quantitative analysis of the evolved volatile compounds by GC-MS. Mirroring the geraniol glucoside concentration of the marc extracts, the monoterpene concentrations following hydrolysis were highest in the Muscat varieties (For details see Appendix A). The Muscat Gordo extract yielded nearly 3600 μg/L of linalool and α-terpineol combined, and Muscat a Petits Grains Blanc (Muscat a PGB) gave approximately 1200 μg/L. The lowest of the floral varieties, Viognier (209 μg/L), gave slightly higher monoterpene levels than the most potent non-floral variety (Sauvignon Blanc, 191 μg/L) while no other non-floral variety yielded more than 120 μg/L of linalool and α-terpineol combined.

### 2.4. Wine Storage of Varietal Marc Extracts

The marc extracts detailed above, plus the same Gewürztraminer marc extract prepared and used in the original work [17], were separately added into a commercially available Chardonnay base wine (pH 3.37), with all extracts added at 0.4 g/L regardless of their expected aromatic potential. The wines were bottled in triplicate and stored for six months to assess the evolution of volatiles from the different marc varieties. Firstly, the geraniol glucoside concentration was determined to observe how these wines aligned with previous observations of increasing geraniol glucoside concentration after the first few months in wine. In all cases, the geraniol glucoside concentration in the wines was higher after six months than the theoretical concentration that was added to the wines based on the extract content of geraniol glucoside (Figure 3). When considered on a ‘percentage increase’ basis (from the expected concentration to the concentration present after six months), there was a trend for an approximate 150–200% increase for the floral varieties. For the non-volatile varieties, there was much more variation, and the absence of geraniol glucoside in the Chardonnay extract prevented a percentage increase comparison across all varieties. This additional variation may be due to significantly lower geraniol glucoside concentrations in the non-floral varieties compared with the floral, and therefore having a greater relative analytical uncertainty. However, even for the non-floral varieties where there were relatively small quantities of geraniol glucoside added, there was again an obvious increase in the geraniol glucoside concentration within the first six months of storage.

With respect to the volatiles evolved over six months of storage, including the potent norsiprenoids β-ionone and β-damascenone, the two Muscat extracts (Muscat Gordo and Muscat a Petits Grains Blanc) provided a much larger increase in total volatile concentration, with an aggregated volatile concentration of over 3500 μg/L and 1000 μg/L, respectively, for the volatiles quantified (Table 2).

All wines with added extracts prepared from floral varieties had a final volatile concentration of above 100 μg/L (Riesling 350; Viognier, 304; 2016 Gewürztraminer 102 μg/L), except for the small-scale Gewürztraminer (53 μg/L). For the non-floral varieties, only the volatile evolution of the Verdelho wine (62 μg/L) exceeded the lowest floral sample, with the remainder of the wines at or below 40 μg/L. However, only the wines with added Muscat Gordo, Muscat a PGB, Riesling and Viognier extracts were significantly different from the control wines. These four extracts also provided statistically significant increases in linalool and α-terpineol, and the two Muscat extracts provided increases in geraniol and nerol. For the non-floral extracts, increases in β-damascenone were the only statistically significant compositional change of the compounds shown in Table 2, with the exception of the wine with added Verdelho marc extract which yielded no significant differences from the control wine.

Analysis of variance between the wines with floral and non-floral additions showed a significant difference by marc extract type (ANOVA output in Appendix A). This was particularly evident for the most abundant monoterpenes, linalool, geraniol and α-terpineol. As such, the volatile evolution follows the expectations created by the designation of varieties into floral and non-floral, to some extent.

When the mean concentrations of the volatiles for each extract were subjected to principal component analysis (PCA), the close positive correlation of the monoterpenes geraniol, nerol, α-terpineol, terpinolene, and, to a lesser extent, β-citronellol is clearly evident, with the Muscat Gordo addition having a particularly high concentration of these compounds, as shown in Table 2. Separated along PC1, the Muscat a PGB and Riesling extracts were moderately high in the compounds, while the non-floral extracts and the Gewurztraminer samples were lower. The Riesling and Muscat Gordo extracts were also high in the norisoprenoids vitispirane, β-damascenone and β-ionone, with those extracts, mainly the non-floral samples, plotted to the upper left quadrant of Figure 4A, also relatively high in the nor-isoprenoids. The Muscat a PGB and the Gewurtztraminer extracts were separated from the other samples on the basis of being higher in the rose oxide compounds.

To remove the impact of the high concentration of volatiles evolving in the wine with added Muscat Gordo marc extract, the relative proportions of volatiles were calculated (Appendix A) and visualised by PCA (Figure 4B). This was performed to see if in wine production, bulking grape marc types together would provide similar volatile profiles, regardless of the magnitude of flavour provided. PC1 accounted for nearly 45% of the variation in the compositional profiles and was the main driver of separation between the floral varieties on one side, and the non-floral on the other. PC1 was defined by the monoterpenes, nerol, geraniol, linalool and α-terpineol, at the negative end, and higher proportions of C_13_-norisoprenoids at the positive end. PC2 was largely defined by the relative proportions of *cis*- and *trans*-rose oxide and correlated with the wine with added laboratory-scale Gewürztraminer marc extract.

## 3. Discussion

The increase in geraniol glucoside over five months of storage to peak at 195% of that in Muscat Gordo extract prior to addition (Figure 1) confirms that found in a previous study of Gewürztraminer marc extract [17]. Similar increases in geraniol glucoside were also observed in the floral varieties over six months of storage (Figure 3), all in the vicinity of a 150–200% increase. For the non-floral varieties, it appears as though increases in geraniol glucoside also occur, but the magnitude was harder to determine as some of the initial geraniol glucoside quantitation in the extracts were below the method limit of quantitation. However, in some instances, the increase calculated was more than 200%.

It is reasonable to assume that the observed increases of geraniol glucoside in all the marc extract spiked wines result from the hydrolysis of disaccharide forms or similar larger molecules such as a malonylated derivative [27,28]. In the analysis of monoterpene RGs and PGs in the Muscat Gordo extract spiked wine (Figure 1, above), the concentrations obtained did not provide a large enough decrease over the course of storage to yield the increases in geraniol glucoside. An improvement in the analytical method to use a more structurally relevant internal standard may be required in the future to better understand the source of the evolved geraniol glucoside. In one previous semi-quantitative survey of monoterpene conjugates, a high prevalence of disaccharide and malonic acid derivatives have been observed [27], although undertaken on grapes rather than on marc extracts. Here, the analysis of juice produced during marc sample preparation also showed a high proportion of monoterpene disaccharides which aligns with that noted in the previous studies (Figure 2F). The lower proportions of monoterpene RGs and PGs observed in the marc extract analysis suggest that an alternative precursor to geraniol glucoside should be sought. Regardless, the analysis of geraniol glucoside content in the marc extracts provided some correlation with the concentration of the major monoterpenes after storage in wines (r = 0.827, Appendix A), while the monoterpenes following acidic hydrolysis of the extracts provided a high correlation with the total concentration of wine monoterpenes (r = 0.996).

In terms of monoterpene composition, acid hydrolysis (pH 1, 100 °C) of the marc extracts to yield monoterpenes gave only two of the major monoterpenes (linalool and α-terpineol) compared with four from storage in wine (geraniol and nerol, additionally). As such, harsh acid hydrolysis of the extracts was not useful for estimating the monoterpene profile that would occur under wine-like storage conditions. The monoterpene profiles from hydrolysis reflected that previously observed for harsh conditions whereby geraniol and nerol are known to be less prevalent under acidic hydrolysis than enzyme hydrolysis [31], or when higher temperatures are used [43]. Analogous to the findings of Loscos et al. (2010), hydrolytic conditions also produced compounds that were not expected to form under wine-like storage, such as TDN being observed in the hydrolysates of all varieties (Appendix A). In short, quantifying volatile evolution from extracts under hydrolytic conditions provided a mechanism to rank extracts with respect to the potency of the extract in wine-like storage conditions but was not useful for predicting the formation of some compounds, such as for TDN as noted previously [44], and extends to understanding the monoterpene composition. However, hydrolysis could be used to rank these extracts in terms of the relative monoterpene evolution during bottle storage, outside of inherent varietal classification depending on the expectation of ‘high’ or ‘low’ monoterpene content.

Previous classification of varieties as either ‘Muscat’, ‘non-Muscat aromatic’ or ‘independent of monoterpenes for flavour’ [1], was altered here to combine the first two categories into ‘floral’ and classify the last as ‘non-floral’. In the initial classification, Viognier was not considered but was later included in an update by Mateo and Jimenez (2000) under ‘Neutral varieties’ [3]. More recently, the importance of monoterpenes for Viognier aroma has been highlighted [45] and one survey of monoterpene concentration by grape variety observed a higher average concentration for Viognier wines (*n* = 20) than for Riesling wines (*n* = 19) [46], which has consistently been considered as ‘floral’ [1,3]. Accordingly, here it was initially categorized as ‘floral’ and the results support that categorization. The analysis of evolved monoterpenes in the wines with added ‘floral’ grape marc extracts after months of storage showed linalool concentrations greater than the odor detection threshold [36], with the two extracts from Muscat varieties also yielding geraniol and α-terpineol above their respective thresholds [36,37]. While no sensory assessment was made in this work, the previous experiment outlined by Parker et al. (2019) showed a clear sensory impact of Gewürztraminer extract additions after six months of storage in either Riesling or Chardonnay wines, which yielded monoterpene concentrations much lower than the Muscat varieties outlined here [17]. As such, it is expected that the addition of the Muscat extracts would provide wines with enhanced flavour, but this would need to be confirmed in future work. While Sauvignon Blanc is ‘non-floral’ with respect to monoterpene content, it has potent and distinctive varietal characteristics due to the presence of ‘tropical’ sulfanyl compounds [47,48]. Previously, Sauvignon Blanc marc has been used as a source for 3-sulfanylhexan-1-ol (3SH) evolution [18]. Here, 3SH was not considered due to the reliance on fermentation for release from amino acid conjugates [49]. However, if future attempts involve the addition of marc extracts prior to fermentation, then the role of 3SH in the aromatic outcome should be considered.

With respect to the logistics of sourcing grape marc for the purpose of producing a latent flavour extract, Muscat varieties will clearly yield potent extracts. Ideally, marc should be kept varietally pure, or at least the Muscat varieties should be kept separately. However, the floral varieties investigated here produced similar volatile profiles after six months of storage in wine, so if only considering the flavour profile rather than the magnitude of flavour, then segregating floral varieties from non-floral may be an option. In terms of non-floral varieties, the relative proportion of C_13_-norisoprenoids evolving during bottle storage was higher than for floral varieties but mainly due to the absence of monoterpenes. As such, while non-floral varieties may not be useful for producing monoterpene-rich latent flavour extracts, they may not alter the profile of an aggregated extract due to the comparatively high monoterpene content in Muscat and other floral varieties.

The extracts produced from red varieties (Shiraz and Cabernet Sauvignon) did yield some colour in addition to wine. In this work, the red marc samples were produced in the same manner as the white marc samples and did not undergo any further extraction. It is known that the extractive nature of red winemaking results in marc samples with significantly lower polyphenolics than for white marc samples [50]. While highly extracted red marc samples would contain less extractable colour, it is expected that they contain significantly lower amounts of extractable flavour also. As such, it is likely that the production of latent flavour extracts should remain the domain of white marc.

Apart from the logistics of sourcing appropriate marc parcels to create an extract rich in monoterpene precursors, the resultant extracts should ideally maintain their potency from production to use. To this end, the Gewürztraminer extract that was prepared in vintage 2016 and detailed by Parker et al. (2019) was used again here after approximately four years of storage at −18 °C. In the initial experiment, when this extract was added to Chardonnay wine (designated C.W in that study and originally stored at pH 3.40), the geraniol glucoside concentration was 2210 μg/L after six months, with geraniol, linalool, nerol and α-terpineol at 19, 45, 3.4 and 33 μg/L, respectively. Here, the same extract yielded concentrations of 2046, 13, 41, 5 and 31 μg/L for the same five compounds (pH 3.37). This result suggests that marc extracts for flavour additions are relatively stable over several years of appropriate storage, providing similar volatile profiles and a similar increase in wine geraniol glucoside of around 200% compared to what was present in the extract.

## 4. Materials and Methods

### 4.1. Muscat Gordo Marc Extraction

Muscat Gordo Blanco marc was collected from a commercial winery in the vintage of 2019 (Riverland, South Australia). The marc was pressed off at the winery and within 5 h was stored at 0 °C. The following day the marc was shoveled into 3 × 240 L drums and stored at −18 °C for 6 months. Before extraction, the marc was allowed to thaw and warm to room temperature before being split into two parcels (150 kg each) and each parcel mixed with 300 L of water in an upright half-tonne fermenter with the temperature held between 10 and 15 °C. The marc/water was left overnight to hydrate, then received regular plunging the following day before being left overnight. Each fermenter was drained and pressed, and the resulting liquid was passed through a cross-flow filter to yield the aqueous extract. This extract was purified in the same manner as described previously [17]. In brief, 50 L aliquots of the extract were loaded onto 3 kg of FPX66 resin in a 6 L column housing, rinsed with caustic (20 L, 25 mL of liquid concentrate into 20 L water—theoretically pH 12.5), water (20 L) and eluted with ethanol (20 L, Tarac, food-grade, spirit neutral). The combined ethanol extracts were concentrated using a high vacuum distillation (VA Filtration/Memstar) to 20 L before being further concentrated using a laboratory rotary evaporator and lyophilised to dryness. This yielded 145.55 g of extract.

### 4.2. Muscat Gordo Extract Hydrolysis over Five-Months in Wine

A commercial Chardonnay was purchased (Yalumba Wine Smiths, 14 × 2 L casks, 13.3% alcohol, pH 3.37, 3.3 g/L of residual sugar, titratable acidity at pH 8.2 6.0 g/L, malic acid 1.85 g/L) and mixed in a 30 L stainless steel keg under a nitrogen atmosphere. The sulfur was adjusted from 30 mg/L to 45 mg/L using a 10% potassium metabisulfite solution. For the control and glycoside addition treatment, approximately 6 L of wine (5.93 kg of 0.9885 g/mL wine) was dispensed into a 9.5 L stainless steel keg pre-filled with carbon dioxide. For the glycoside addition, 2.4 g of marc extract, pre-dissolved in the wine was added and gently mixed (0.4 g/L in wine). The wine was dispensed into 15 × 375 mL brown glass bottles pre-filled with carbon dioxide and then sealed with crown seals. The control was handled in the same way, without the glycoside addition. The wines were stored at 15 °C for six months, with three bottles opened every month for analysis of volatiles by GC-MS and non-volatiles by LC-MS/MS as detailed below.

### 4.3. Small-Scale Marc Generation: Grapes and Processing

Grape varieties were classified as either floral or non-floral, based on their monoterpene profiles similar to that described in Strauss et al. (1986), and hand-picked approximately at commercial maturity (20–24 Brix for whites, 23–26 for reds). In 2017, several Muscat and non-Muscat floral varieties were collected from different locations (Muscat a Petits Grains Blanc [Muscat a PGB] and Viognier from McLaren Vale, South Australia; Riesling, Muscat Gordo and Gewürztraminer from Barossa Valley, South Australia) and in 2018, several non-floral varieties were collected from the Barossa Valley (Chardonnay, Semillon, Verdelho, Sauvignon Blanc, Shiraz and Cabernet Sauvignon). Approximately 10 kg of grapes were hand-picked at commercial harvest and stored at 4 °C for a maximum of two days before being processed. A known mass of grapes was divided into 2 kg parcels and each parcel was separately crushed and pressed in a 2 kg capacity stainless steel hand press. Each parcel was pressed three times to 20 Nm using a torque wrench, and between press cycles, the press cake was broken up and mixed to ensure even pressing of all bunches. The juice and marc from the entire 10 kg sample were pooled and weighed, then stored at −18 °C and retained for later analysis. The marc weight as a percentage of grape weight crushed for each variety was: Muscat a Petits Grains Blanc, 45.6%; Viognier, 53.2%; Riesling, 51.0%; Muscat Gordo, 56.6%; Gewürztraminer, 49.0%; Chardonnay, 45.8%; Semillon, 44.4%; Verdelho, 46.3%; Sauvignon Blanc, 41.4%; Shiraz, 47.4%; Cabernet Sauvignon, 51.7%.

### 4.4. Small-Scale Marc Extraction

For each variety, the marc was defrosted and extracted in water (1.5 L per kg of marc) containing potassium metabisulfite (150 mg/L) with constant stirring for 24 h at ambient temperature. The extract was strained to remove the marc then centrifuged (Beckman, 9000 RPM, 10 min) before being purified. The extract was loaded onto FPX66 resin, washed with caustic solution (2 L, 1.26 g/L), then water (2 L), and eluted with ethanol (2 L). The ethanolic extract was concentrated using a rotary evaporator then lyophilised to yield the purified marc extract. For full details of grape pressing and extract production outcomes, see Appendix A.

### 4.5. Small-Scale Marc Extract Wine Storage Trial

The storage trial was conducted using the same 30 L keg of commercial wine as described above. For each treatment, approximately 1.2 L of wine (1.186 kg of 0.9885 g/mL wine) was dispensed into a 2 L stainless steel keg pre-filled with carbon dioxide followed by the addition of 0.4 g/L of marc extract, pre-dissolved in a small volume of the wine. After gentle mixing, the wine was dispensed into 3 × 375 mL brown glass bottles pre-filled with carbon dioxide and then sealed with crown seals. The wines were stored at 15 °C for six months before being assessed chemically.

### 4.6. Chemical Analysis of Extracts and Wines

Quantitation of geraniol glucoside by LC-MS/MS using a d_2_-geraniol glucoside internal standard was as previously reported [17] and *d*_3_-syringol gentiobioside was synthesized in-house as previously described [51]. Wines were analysed neat and grape marc extracts were dissolved in water, followed by the addition of the internal standards (d_2_-geraniol glucoside for quantification of geraniol glucoside, and *d*_3_-syringol gentiobioside for disaccharide bound monoterpenes) prior to instrumental analysis. Transitions monitored: syringol gentiobioside (537 → 323, 537 → 477), *d*_3_-syringol gentiobioside (540 → 323, 540 → 341), monoterpene pentosyl-glucose (507 → 293, 507 → 311, 507 → 477) and monoterpene rhmanosyl-glucose (521 → 307, 521 → 325, 521 → 461).

The volatile potential of grape marc extracts was performed as per Grebneva et al. (2019), where a weighed amount was dissolved in water, acidified to pH 1, and heated at 100 °C for an hour before the hydrolysate was cooled prior to volatiles analysis [44]. The volatiles present in the extract hydrolysates or wine samples for a range of monoterpenes and C_13_-norisoprenoids by SIDA SPME-GC-MS, as previously described [17].

Volatile evolution in wine was determined for monoterpenes and C_13_-norisoprenoids as described below. Upon opening, each wine replicate was analysed separately by GC-MS to determine the volatile evolution. Glycosidic analysis was performed via LC-MS/MS using deuterated-geraniol glucoside as an internal standard for quantification of geraniol glucoside, and deuterated syringol gentiobioside as an internal standard for disaccharide bound aglycones.

### 4.7. Membrane-Assisted Solvent Extraction GC-MS Method

Deuterium labelled compounds for the stable isotope dilution assay (SIDA) were synthesised in-house as previously reported: *d*_6_-1,8-cineole and *d*_6_-α-terpineol [52]; *d*_7_-geraniol [53]; *d*_6_-linalool [54]; *d*_2_-β-citronellol [55]; *d*_4_-β-damascenone, *d*_3_-α-ionone and *d*_3_-β-ionone [56]. Monoterpenes and C_13_-norisopernoids were analysed using membrane assisted solvent extraction-gas chromatography-mass spectrometry (MASE-GC-MS) on an Agilent 7890B GC (Agilent Technologies Australia Pty Ltd., Mulgrave, VIC, Australia), coupled to an Agilent 5977B MS and equipped with a Gerstel MPS Robotic Pro (Lasersan Australasia Pty Ltd., Tanunda, SA, Australia). The GC was fitted with a Deans switch (Agilent) to utilise a post-run backflush program. Samples were prepared in a 20 mL glass crimp-cap, headspace vial (Gerstel, Lasersan Australasia) containing a magnetic stir bar (3 × 8 mm) by addition of sample (15 mL), ethanol solution containing labelled compounds as the internal standards (50 μL, equivalent to a ~17 μg/L addition of each *d*_6_-1,8-cineole, *d*_6_-linalool, *d*_6_-α-terpineol, *d*_2_-β-citronellol, *d*_7_-geraniol, *d*_4_-β-damascenone, *d*_3_-α-ionone, and *d*_3_-β-ionone), and MASE apparatus (membrane bag, sealing ring, and cone insert; Gerstel), then crimp capped and placed on a cooler tray held at 10 °C. Immediately prior to analysis, the MPS Robotic Pro added hexane/acetone (2:1, 0.9 mL) into the membrane bag, the vial was stirred for 45 min at 35 °C, cooled for 15 min at 10 °C, then 2 µL of the extract was injected. The inlet was set at 250 °C in splitless mode with a constant pressure of 43.7 psi and lined with an ultra-inert glass liner with glass wool for liquid injections (Agilent, 6.5 mm o.d., 4.0 mm i.d., 78.5 mm long); the purge valve was opened after 2 min at a split ratio of 25:1. Between the inlet and the Deans Switch a VF-35 ms (60 m × 0.25 mm i.d. with 0.25 μm film thickness) was installed with the carrier gas, helium (ultra-high purity, BOC, Adelaide, SA, Australia), held at a constant pressure rate of 24.2 psi. Installed between the Deans Switch and the MS was a restrictor column of deactivated silica (5 m × 0.15 mm i.d.). The oven temperature began at 40 °C and held for 1 min, before being increased at 20 °C/min to 100 °C, then increased at 5 °C/min to 225 °C. At 29 min, after all target peaks were eluted, the column was backflushed (−3.55 mL/min) for 4 min at 280 °C. The transfer line was held at a constant temperature of 280 °C. The instrument was controlled with Agilent MassHunter software (B.07.06.2704) in conjunction with Gerstel Maestro Software (Version 1.5.3.67). Data analysis was performed in MassHunter (Agilent, Version B.09.00). The ions monitored for the internal standards (ion used for quantitation is underlined, % in parentheses) were: *d*_6_-1,8-cineole, 160, 142 (72), 113 (67); *d*_6_-linalool, 124, 142 (75), 93 (350); *d*_6_-α-terpineol, 142, 124 (95), 65 (79); *d*_2_-β-citronellol, 158, 125 (281), 69 (3850); *d*_7_-geraniol, 128, 99 (104), 75 (150); *d*_4_-β-damascenone, 194, 179 (278), 73 (27); *d*_3_-α-ionone, 195, 139.1 (219), 124 (219); *d*_3_-β-ionone, 180, 195 (5), 46.1 (13). For method validation, high and low concentration spikes (*n* = 7 at each concentration) were performed to provide compound recovery and repeatability. The spikes were made at 50.0 and 5.0 μg/L for limonene, 1,8-cineole, terpinolene, linalool, α-terpineol, β-citronellol, nerol and geraniol, 4.0 and 0.2 μg/L for *cis*-rose oxide, 1.5 and 0.1 μg/L for *trans*-rose oxide, 10.0 and 0.5 μg/L for vitispiranes, TDN, β-damascenone, α-ionone and β-ionone, and 6.0 and 0.3 μg/L for wine lactone.

### 4.8. Data, Graphing and Statistics

All data were handled using R [57] via the RStudio IDE (version 1.3.1093 “Apricot Nasturtium”) [58]. Specifically, functions within the ‘stats’ package were used for the analysis of variance (aov), post-Hoc analysis (TukeyHSD), Shapiro-Wilk test (shapiro.test), Wilcoxon rank-sum test (wilcox.test) and principal component analysis (prcomp). Most graphs were produced using Tidyverse [59], while Principal Component Analysis was visualized using Factoextra [60].

## 5. Conclusions

Here, the developed membrane-assisted solvent extraction (MASE) GC-MS method could quantify key wine monoterpenes and C_13_-norisoprenoids in a single method and at concentrations below their respective odor detection thresholds in both white and red wines. When coupled with LC-MS/MS quantitation of monoterpene glycosides, these analytical methods provided some insight into the hydrolysis pathway to yield monoterpenes from marc-derived extracts. Specifically, an increase in geraniol glucoside was consistently observed after five to six months of storage in wine, although the decreases in disaccharide-bound monoterpenes were not large enough to confirm them as the source of the evolved geraniol glucoside, and more work is required to completely elucidate the monoterpene evolution pathway in these extracts.

The assessment of the grape marc extracts for potency using either analysis of geraniol glucoside via LC-MS/MS or by hydrolysis and subsequent GC-MS analysis of the evolved volatiles proved adequate, even with the subsequently observed increases in geraniol glucoside. However, the volatile analysis after hydrolysis of the extracts did not provide useful information as to the profile of different monoterpenes, only the total magnitude.

The addition of extracts to wine that were derived from varieties determined as ‘floral’ provided a greater increase of monoterpenes than the ‘non-floral’ extracts, although, of the ‘floral’ varieties, the Muscat-derived extracts were clearly more potent. As such, the logistics of obtaining grape marc as an input to produce latent flavour extracts are complicated by the benefit of maintaining varietal separation as much as is possible, or at least by removing ‘floral’ varieties from other white varieties and red grape marc. Furthermore, the comparison of a Gewürztraminer extract when it was produced and again after four years of storage suggests that these extracts are amenable to storage and use across multiple vintages.

## Figures and Tables

**Figure 1 molecules-27-02046-f001:**
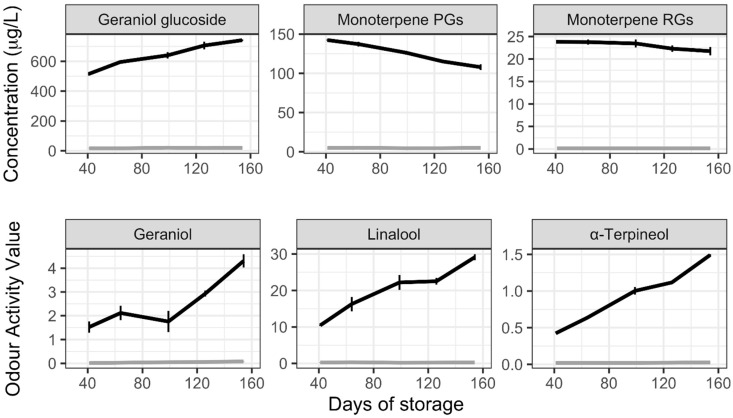
Change in concentration of glycosidically bound monoterpenes and odour activity value (OAV) of free monoterpenes in Chardonnay wines over 156-days of storage in a bottle with (black lines) and without (grey lines) 0.4 g/L addition of Muscat Gordo marc extract. Error bars represent the standard deviation of three wines. PG, pentosyl-glucose; RG, rhamnosyl-glucose. OAVs were determined using thresholds of 30 μg/L for geraniol, 25 μg/L for linalool and 250 μg/L for α-terpineol [36,37].

**Figure 2 molecules-27-02046-f002:**
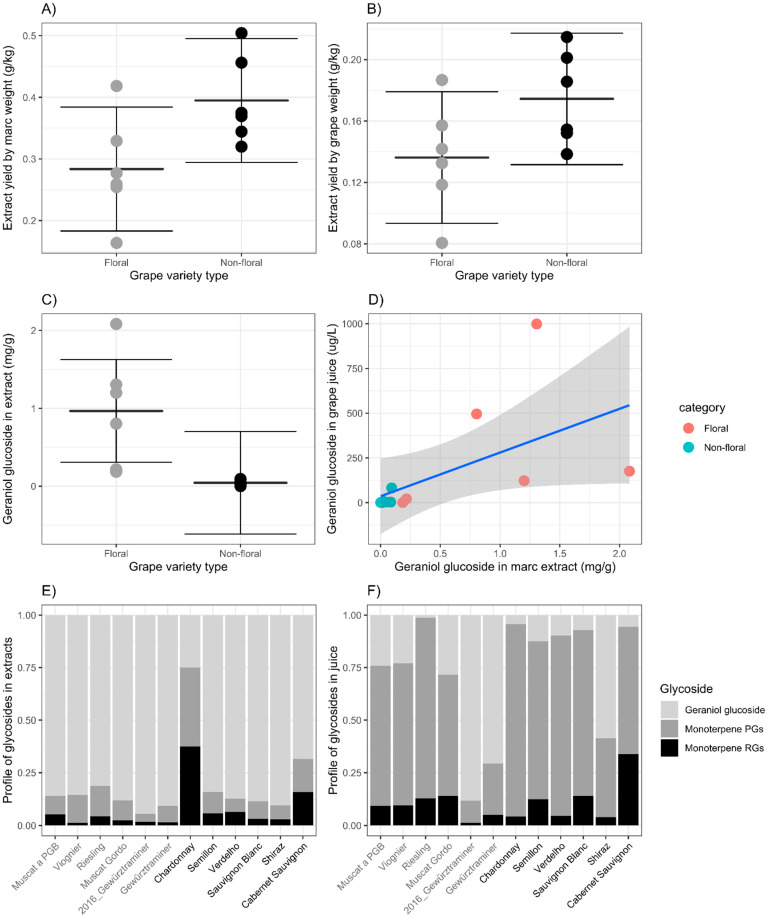
Laboratory-scale extraction of varietally pure grape marc samples separated into floral (grey points or labels) or non-floral (black points or labels). (**A**) Yield of extract per mass of marc extracted, separated by floral or non-floral varieties. (**B**) Yield of extract per mass of grapes processed, separated by floral or non-floral varieties. (**C**) Geranyl glucoside concentration in extracts, separated by floral or non-floral varieties. (**D**) Comparison of geranyl glucoside concentration in marc extract versus in grape juice. (**E**) Profiles of monoterpene glycosides in grape marc extracts. (**F**) Profiles of monoterpene glycosides in the juice of each variety. Where present, error bars extend to Tukey HSD at 95%.

**Figure 3 molecules-27-02046-f003:**
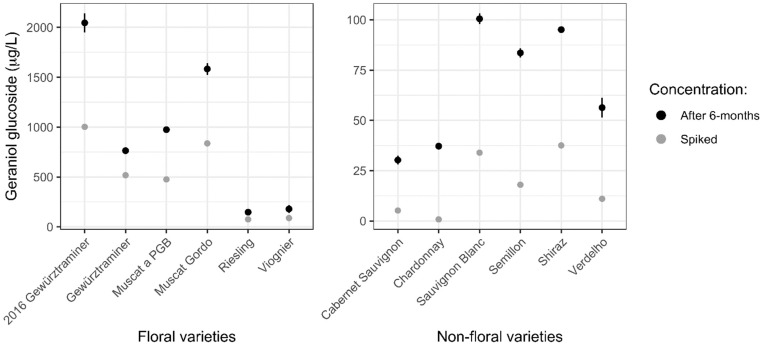
Concentration of geraniol glucoside in a Chardonnay base wine with 0.4 g/L of added marc extract from different varieties, showing the mean concentration and standard deviation of wine triplicates after six months of storage (black) and the theoretical amount added to the base wine based on the geraniol glucoside concentration in the extracts (grey).

**Figure 4 molecules-27-02046-f004:**
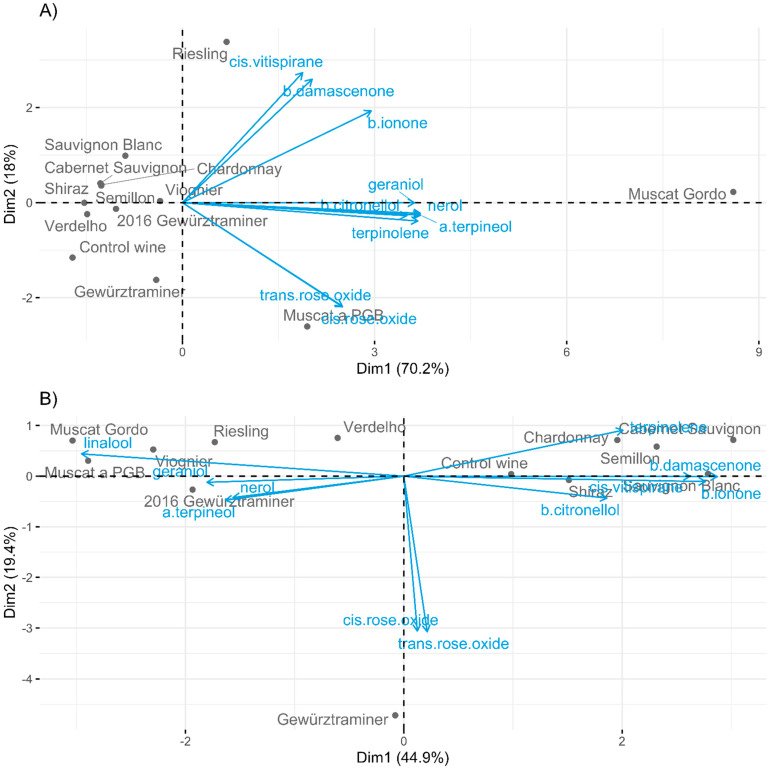
PCA biplots of the volatile compounds present in Chardonnay wines spiked with marc extracts from different varieties, analysed after six months of storage, using the mean of triplicate wines. (**A**) Absolute concentration of volatiles in wines. (**B**) Volatiles as a proportion of the total volatiles quantitated in the analytical suite.

**Table 1 molecules-27-02046-t001:** Calibration and validation data for the membrane-assisted solvent extraction (MASE) GC-MS method in a white wine matrix. High and low spike concentrations are detailed in the Materials and Methods.

Analyte	Retention Time (Mins)	Internal Standard	Ions Monitored (*m*/*z*)		Linearity		Limits (µg/L)		Aroma Detection Threshold		Low Spike		High Spike	
			Quantifier	Qualifiers (%)	R^2^	Range (µg/L)	Detection	Quantitation	Concentration (µg/L)	Reference	Recovery (%)	Repeatability (RSD %)	Recovery (%)	Repeatability (RSD %)
Limonene	13.86	*d*_6 _-1,8-Cineole	136	93 (230), 68 (255)	0.9891	0, 0.47–505	0.03	0.09	10	[34]	110.8	8.6	96.9	8.1
1,8-Cineole	14.41		111	154 (13), 126 (128)	0.9985	0, 0.47–500	0.04	0.12	3.2	[35]	100.8	2.7	101.1	1.6
Terpinolene	15.55	*d*_6_-Linalool	136	121 (106), 105 (28)	0.9838	0, 0.47–500	0.08	0.27	-	-	108.5	8.3	95.3	9.2
Linalool	15.73		71	136 (40), 121 (19)	0.9981	0, 4.80–510	1.19	3.95	25	[36]	98.8	5.0	101.3	1.8
(-)-*cis*-Rose oxide	16.18		139	154 (14), 140 (10)	0.9909	0, 0.39–416	0.01	0.03	0.2	[37]	96.0	11.3	102.2	8.2
(-)-*trans*-Rose oxide	16.8		139	154 (10), 140 (11)	0.9856	0, 0.15–155	0.01	0.02	160	[38]	98.0	10.8	106.6	10.3
α-Terpineol	19.41	*d*_6_-α-Terpineol	136	93 (110), 59 (114)	0.9987	0, 0.47–500	0.09	0.30	250	[36]	94.7	5.0	97.1	2.6
β-Citronellol	19.49	*d*_ 2 _-β-Citronellol	123	156 (38), 69 (440)	0.9973	0, 4.84–518	0.92	3.07	700	[36]	104.5	3.8	98.3	2.7
Nerol	19.85		139	154 (48), 121 (4200)	0.9924	0, 2.08–519	0.62	2.05	700	[36]	96.9	10.0	104.8	5.6
Geraniol	20.62	*d*_7_-Geraniol	136	93 (440), 123 (400)	0.9921	0, 4.67–500	0.68	2.25	30	[37]	91.4	4.8	95.7	1.8
Vitispiranes	18.22	*d*_8_-Naphthalene	192	177 (61)	0.9957	0, 0.05–108	0.02	0.05	101	[39]	101.8	6.5	111.2	7.6
TDN	21.22		157	172 (31), 142 (45)	0.9932	0, 0.11–105	0.03	0.09	2	[40]	81.4	10.9	103.7	7.0
β-Damascenone	25.43	*d*_4_-β-Damascenone	69	190 (36), 175 (16)	0.9981	0, 0.53–106	0.08	0.27	0.05	[37]	84.9	5.2	99.0	2.4
α-Ionone	26.21	*d*_3_-α-Ionone	136	121 (125), 93 (210)	0.9985	0, 0.11–108	0.02	0.06	2.6	[41]	100.3	1.7	99.1	1.3
β-Ionone	28.15	*d*_3_-β-Ionone	177.1	192 (13), 43 (6)	0.9985	0, 0.05–107	0.01	0.02	0.09	[42]	80.0	1.6	99.8	3.9
Wine lactone	26.20	*d *_3_-Wine lactone	151	166 (27), 138 (14)	0.9939	0, 2.01–201	0.52	1.72	0.01	[37]	109.3	5.5	102.7	2.6

**Table 2 molecules-27-02046-t002:** Concentration (in μg/L) of selected monoterpenes and C_13_-norisoprenoids in control Chardonnay wine and Chardonnay base wine with 0.4 g/L additions of marc extract from different varieties, analysed after six months of storage at 15 °C. ‘Total’ represents the sum of the volatiles in the analytical suite, including those shown in Table A1 (Appendix B). Data expressed as mean of triplicates ± standard deviation, with the Tukey 95% honest significant difference (HSD) for each compound.

Sample	Limonene	Linalool	α-Terpineol	β-Citronellol	Nerol	Geraniol	β-Damascenone	β-Ionone	*cis*-Rose oxide	Total
Control wine	0.14 ± 0.04 ^+^	5.92 ± 0.21	5.67 ± 0.04	4.17 ± 0.00	1.02 ± 0.00 ^#^	2.08 ± 0.11	1.35 ± 0.07	0.03 ± 0.00	0.01 ± 0.00 ^+^	21.08 ± 0.01
2016 Gewurztraminer	0.16 ± 0.05	40.80 ± 1.24	30.65 ± 1.10	3.00 ± 0.25	5.01 ± 0.72	13.21 ± 0.84	7.34 ± 0.24	0.03 ± 0.01	0.15 ± 0.03	102.24 ± 1.51
Muscat a PGB	1.04 ± 0.23	560.68 ± 14.72	336.62 ± 14.52	3.72 ± 0.40	43.06 ± 1.4	99.10 ± 4.23	7.40 ± 0.33	0.03 ± 0.02	0.75 ± 0.05	1055.25 ± 23.26
Viognier	0.37 ± 0.02	158.29 ± 40.71	94.91 ± 24.74	3.89 ± 0.99	10.65 ± 2.43	28.21 ± 5.80	6.76 ± 1.79	0.10 ± 0.02	0.11 ± 0.03	304.89 ± 75.95
Riesling	0.33 ± 0.08	177.61 ± 26.38	110.72 ± 13.97	2.77 ± 0.01	10.82 ± 4.73	26.99 ± 9.00	17.46 ± 3.00	0.14 ± 0.02	0.04 ± 0.02	351.12 ± 57.40
Muscat Gordo	3.21 ± 0.35	1743.91 ± 16.17	936.46 ± 37.74	11.90 ± 0.54	144.39 ± 12.25	760.18 ± 69.52	15.50 ± 0.71	0.19 ± 0.00	0.56 ± 0.02	3622.41 ± 107.13
Gerwurztraminer	0.14 ± 0.06 ^#^	17.08 ± 0.19	16.36 ± 0.24	3.38 ± 0.67	1.95 ± 0.13	5.44 ± 0.14	6.94 ± 0.48	0.04 ± 0.01	0.53 ± 0.05	53.31 ± 0.71
Chardonnay	0.14 ± 0.05 ^#^	11.88 ± 0.41	11.43 ± 0.20	2.08 ± 0.66 ^+^	1.02 ± 0.00 ^##^	3.31 ± 0.17	9.52 ± 0.41	0.06 ± 0.00	0.03 ± 0.00	40.90 ± 1.19
Semillon	0.11 ± 0.00 ^##^	8.99 ± 0.37	10.37 ± 0.56	2.69 ± 0.29	1.02 ± 0.00 ^##^	2.90 ± 0.2.00	8.23 ± 0.88	0.04 ± 0.01	0.03 ± 0.01	36.30 ± 2.13
Verdelho	0.21 ± 0.01	27.91 ± 2.12	21.72 ± 1.01	1.71 ± 0.68 ^#^	2.61 ± 0.36	4.65 ± 0.69	3.37 ± 0.30	0.05 ± 0.00	0.03 ± 0.00	64.04 ± 3.09
Sauvignon Blanc	0.11 ± 0.00 ^##^	9.20 ± 0.31	10.96 ± 0.23	2.61 ± 0.39	1.02 ± 0.00 ^##^	3.09 ± 0.07	12.86 ± 0.56	0.07 ± 0.02	0.04 ± 0.00	41.84 ± 0.77
Shiraz	0.11 ± 0.00 ^##^	7.95 ± 0.15	8.97 ± 0.16	1.80 ± 0.82 ^#^	1.25 ± 0.40 ^#^	1.96 ± 0.69 ^+^	6.74 ± 0.40	0.03 ± 0.00	0.03 ± 0.00	30.49 ± 0.63
Cabernet Sauvignon	0.14 ± 0.06 ^#^	6.01 ± 0.24	8.13 ± 0.13	2.68 ± 0.05	1.02 ± 0.00 ^##^	2.38 ± 0.15	11.15 ± 2.30	0.03 ± 0.01 ^+^	0.02 ± 0.00	33.13 ± 2.47
Tukey 95% HSD	0.38	43.07	41.82	1.67	11.25	60.58	3.39	0.04	0.07	120.40

Plus, double plus, triple plus symbols refer to one (^+^), two (^#^) or three (^##^) replicates with concentrations represented by <LOQ or <LOD replacement values.

## Data Availability

All data are included in the article and/or Appendix A.

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
