# Peer review of "The Varietal Influence of Flavour Precursors from Grape Marc on Monoterpene and C13-Norisoprenoid Profiles in Wine as Determined by Membrane-Assisted Solvent Extraction (MASE) GC-MS"

_molecules, 2022, doi:10.3390/molecules27072046_

Round 1

Reviewer 1 Report

This work made a good try to reuse the grape marc to extract flavour precursors to strengthen the varietal aroma of a wine. The workers employed reliable methods and the data supports the conclusions. However, there are still some aspects the authors could consider to improve their manuscript:

  1. In the abstract section the expression is "In all wines with added marc extract, geraniol glucoside concentration increased by around 150-200%, regardless of cultivar", while for the result section, “there was a trend for an approximate 150-200% increase for the floral varieties. For the non-volatile varieties there was much more variation”. The two statements are inconsistent.
  2. The content and decreasing trend of pentosyl-glucose (PGs) and rhmanosyl-glucose (RGs) are different, and the relationship between these two substances and the formation of geraniol glucoside deserves further discussion. It may be possible to carry out correlation analysis on three types of substances, PGs and RGs, geraniol glucoside and major monoterpenes, to explore the evolution of flavor substances from grape marc.
  3. The Discussion section could be tweaked to make it more logical and focused.This section is a further integration of the experimental results and more in-depth exploration of the scientific questions raised, rather than as a supplement to the Results section, e.g. in page 14 "Gewürztraminer" extract that was prepared in vintage 2016" appears here as an extra note that is not very relevant to the subject of the article.
  4. The authors simply measured the changes in the volatile content of wine after adding the extract, perhaps adding to the discussion of some effects on the macroscopic expressionof aroma characteristics.
  5. Since this study was applied to commercial wine, food safety with the addition of grape pomace extract should also be considered. Does the addition of the extract comply with the regulations on wine additives? I think this issue should be addressed in the article.

Author Response

As per attached file

Reviewer 2 Report

In my opinion results presented in manuscript are interesting, with actual idea and have the potencial of practical use in vine sector. Experiment was planned and done methodologically well, the results are properly described and reflect the goal of research.

I have a few comments and suggestions:

1) it is difficult to understand the Table 1. There are presented experimental data of authors or data from literature sources? What represent numbers [34] [35] [36] and etc. in column „Aroma detection threshold (µg/L)“? Literature source? If yes, I suggest these numbers to place in a separate column at the end of the table. Also, why this column is highlighted in a different colour? I suggest to think and present this Table clearer.

2) the units of measurement for the quantities of compounds must be indicated in Table 2.

3) lines 538-541: why some numbers are underlined? It must be explained.

4) the units of measurement for the quantities of compounds must be indicated in Table 3.

5) I suggest to write a conclusions section.

Author Response

As per attached file
